# Therapeutic Options and Outcomes for the Treatment of Neonates and Preterms with Gram-Negative Multidrug-Resistant Bacteria: A Systematic Review

**DOI:** 10.3390/antibiotics11081088

**Published:** 2022-08-10

**Authors:** Lorenzo Chiusaroli, Cecilia Liberati, Maria Caseti, Luigi Rulli, Elisa Barbieri, Carlo Giaquinto, Daniele Donà

**Affiliations:** Division of Pediatric Infectious Diseases, Department for Women’s and Child’s Health, University of Padua, 35128 Padua, Italy

**Keywords:** MDR bacteria, Gram-negative, Enterobacteriaceae, *A. baumannii*, *P. aeruginosa*, neonates, term, preterm, outcome, treatment, review

## Abstract

(1) Background: Infections caused by multidrug-resistant (MDR) or extensively drug-resistant (XDR) bacteria represent a challenge in the neonatal population due to disease severity and limited therapeutic possibilities compared to adults. The spread of antimicrobial resistance and drug availability differ significantly worldwide. The incidence of MDR bacteria has constantly risen, causing an increase in morbidity, mortality, and healthcare costs in both high-income (HIC) and low- and middle-income countries (LMIC). Therefore, more evidence is needed to define the possible use of newer molecules and to optimize combination regimens for the oldest antimicrobials in neonates. This systematic review aims to identify and critically appraise the current antimicrobial treatment options and the relative outcomes for MDR and XDR Gram-negative bacterial infections in the neonatal population. (2) Methods: A literature search for the treatment of MDR Gram-negative bacterial infections in neonates (term and preterm) was conducted in Embase, MEDLINE, and Cochrane Library. Studies reporting data on single-patient-level outcomes related to a specific antibiotic treatment for MDR Gram-negative bacterial infection in children were included. Studies reporting data from adults and children were included if single-neonate-level information could be identified. We focused our research on four MDROs: *Enterobacterales* producing extended-spectrum beta-lactamase (ESBL) or carbapenemase (CRE), *Pseudomonas aeruginosa,* and *Acinetobacter baumannii*. PROSPERO registration: CRD42022346739 (3) Results: The search identified 11,740 studies (since January 2000), of which 22 fulfilled both the inclusion and exclusion criteria and were included in the analysis. Twenty of these studies were conducted in LMIC. Colistin is the main studied and used molecule to treat Gram-negative MDR bacteria for neonate patients in the last two decades, especially in LMIC, with variable evidence of efficacy. Carbapenems are still the leading antibiotics for ESBL *Enterobacterales*, while newer molecules (i.e., beta-lactam agents/beta-lactamase inhibitor combination) are promising across all analyzed categories, but data are few and limited to HICs. (4) Conclusions: Data about the treatment of Gram-negative MDR bacteria in the neonatal population are heterogeneous and limited mainly to older antimicrobials. Newer drugs are promising but not affordable yet for many LMICs. Therefore, strategies cannot be generalized but will differ according to the country’s epidemiology and resources. More extensive studies are needed to include new antimicrobials and optimize the combination strategies for the older ones.

## 1. Background

The emergence of multidrug-resistant organism (MDRO)-associated infections also represents a global challenge for the neonatal population, and the shortage of newer antimicrobial options is a prioritized global health concern [1,2,3]. Infections contribute considerably to neonatal mortality, with sepsis accounting for 15.6% of neonatal deaths (an estimated 430,000–680,000 deaths per annum), which predominantly occur in low- and middle-income countries (LMICs) [4,5,6]. In high-income countries (HICs), neonates admitted to the neonatal intensive care unit (NICU) are at higher risk of infections [7]. Gram-negative bacteria are responsible for approximately 15–30% of late-onset infections (LOIs) in HIC, especially *E. coli* and *Klebsiella* species [8,9]. In LMIC, on the other hand, Gram-negative bacteria are the most frequently identified neonatal pathogens [10,11], and more than half are resistant to three or more classes of broad-spectrum antibiotics (Penicillin, Cephalosporins, Fluoroquinolones, and Carbapenems) [12].

Further, MDROs are colonizing neonates across NICUs with an increasing trend in both HICs and LMICs [13,14].

In Europe, third-generation cephalosporin resistance in *E. coli* increased up to 50% of strains in 2020, while resistance to carbapenems was as high as 25% in *Enterobacterales* and 50% in Acinetobacter spp.; *Pseudomonas aeruginosa* has gained multiple resistances against common antimicrobials, with 50% of strains resistant to three or more antibacterial categories [15]. In the same way, in the USA, the 2019 Center for Disease Control and Prevention (CDC) report designated extended-spectrum β-lactamase-producing *Enterobacterales* (ESBL-E), carbapenem-resistant *Enterobacterales* (CRE), and *Pseudomonas aeruginosa* with difficult-to-treat resistance (DTR-P.A.) as urgent or serious threats [16]. In LMICs, resistance patterns are even higher, with a rate of carbapenem-resistant *K. pneumoniae* that, in 2019, was reported to be up to 34%, while *Acinetobacter baumannii* was up to 84% [17].

As a result, antibiotics consumption increased worldwide by 65% between 2000 and 2015, mainly older molecules in LMIC; however, polymyxins and tigecycline also increased, with a worrisome trend in high-income countries (HIC) [13,18].

Newer promising agents, such as beta-lactam/beta-lactamase inhibitors (βL-βLI), are studied for resistant Gram-negative infections, but studies in neonatal patients are limited [19]. Further, they are expensive compared to older molecules and are not available worldwide.

This systematic review aims to critically appraise the current antimicrobial treatment options and the relative outcomes for MDR and extensively drug-resistant (XDR) Gram-negative bacterial infections in the neonatal population.

## 2. Methods

### 2.1. Literature Search

This systematic review was carried out according to the Preferred Reporting Items for Systematic Reviews and Meta-Analyses (PRISMA) guidelines (Figure 1). Embase, Medical Literature Analysis and Retrieval System Online (MEDLINE), and Cochrane Library were searched for relevant studies, combining Medical Subject Heading (MeSH) and free-text terms for “neonate” and “MDR” and “Outcome assessment” (complete search strategy in Additional File 2). The search strategy involved restrictions on the date (from 1 January 2000 to 7 September 2021) but not on language. All studies on children younger than 18 were searched.

This study is registered with the International Prospective Register of Systematic Reviews (PROSPERO) as record number CRD42022346739.

We included studies with any method of diagnosing MDR infections in the neonate and preterm populations; however, the reasons for defining an organism as MDR were not explicit in most studies, but we can assume the international expert proposal definition was applied [20]. Any site of infection was included. The search results were exported to Rayyan software for further manuscript assessment and handling.

### 2.2. Study Selection

Assessments of the titles, abstracts, and full texts were conducted independently by three investigators (L.C., M.C., and L.R.). Discussion with a fourth reviewer (D.D.) resolved any disagreement regarding study selection.

### 2.3. Eligibility Criteria

Eligible study designs included randomized clinical trials, observational studies, prospective or retrospective designs, concomitant or historical control studies, case series, and case reports. Meta-analyses, systematic reviews, and narrative reviews were not included. Studies investigating any antimicrobial treatment for infections caused by MDR Gram-negative bacteria were included.

Studies without pediatric data or about Gram-positive bacterial infections or drug resistance on malaria, human immunodeficiency virus (HIV), viral tuberculosis, and fungal treatment were also excluded.

The population of interest was term and preterm newborns with confirmed MDRO Gram-negative infections with clearly defined resistance that were receiving antimicrobial treatment and presenting clinical and microbiological outcomes.

The primary outcome was infection-related mortality from the initiation of treatment until discharge. Secondary outcomes were clinical success (defined as complete resolution or a substantial improvement in the signs and symptoms of the index infection) and microbiological success (measured by the suppression, eradication, or relapses of bacterial growth).

Studies published between 1 January 2000 and 7 September 2021 were included. Further details are reported in the PICOS (P: problem/patient/population; I: intervention; C: comparison/control; O: outcome; S: Study design) (Additional File 1).

### 2.4. Data Extraction and Assessment of Study Quality

The following data were extracted using a standardized data collection form:Study characteristics (authors, year of publication, study design, study location, and country);Patient characteristics (age, care setting, and inclusion and exclusion criteria);Type of MDR;Setting;Main results with accuracy measures;Health outcomes (e.g., mortality, clinical response, and microbiological eradication);Main results.

Standardized predetermined study criteria were applied to all full-text documents. The selection process is presented in Figure 1.

The quality and risk of bias in individual studies were jointly assessed at the study and outcome level by all reviewing authors using the Study Quality Assessment Tool from The National Heart, Lung, and Blood Institute [21]. The quality assessment results are presented in Table 1, Table 2, Table 3 and Table 4.

### 2.5. Summary Measures

The following measures of treatment success were included: absolute values, absolute risk differences, hazard ratio (HR), relative risk, and odds ratio. Unadjusted and adjusted measures were included if available.

## 3. Results

### 3.1. Study Selection

A total of 11,740 records were found, and 2225 duplicate records were removed. Then, 9515 records were screened and excluded by title or abstract, and 164 were excluded after reading the full text because they did not meet the eligibility criteria, and 42 articles regarding Gram-positive bacteria were excluded. A total of twenty-two articles were included in the systematic qualitative review (Figure 1). We report the characteristics of the studies in Table 1, Table 2, Table 3 and Table 4. Figure 2 shows the studies with a relevant number of patients.

### 3.2. ESBL Enterobacterales

We included only one retrospective study regarding ESBL-producing *Enterobacterales* in neonates [22]. The mortality, clinical success, and microbiological eradication are displayed in Table 1. The study was conducted in an LMIC to observe the treatment and related outcomes of ESBL infections in neonates: a case series of 100 neonates infected with ESBL-producing *Klebsiella* (bloodstream infection (BSI) or meningitis) described an overall mortality of 30%, with rates of 25% if empirical therapy was undertaken with piperacillin-tazobactam and amikacin, 32% with meropenem, 42% with no empiric therapy or when other antibiotics were used [22].

### 3.3. Carbapenem-Resistant Enterobacterales (CRE)

We included eight articles regarding CRE (including carbapenemase-producing *K. pneumoniae*-KPC and XDR *Enterobacterales*): three case reports, two case series, and three retrospective studies [23,24,25,26,27,28,39,40]

The mortality and clinical and microbiological outcomes are presented in Table 2. The settings were different, but most studies (6/8) were conducted in LMICs.

The selected studies report the following CRE-associated clinical settings: BSI, ventilator-associated pneumonia (VAP), meningitis, and others.

The included antibiotic regimens were colistin (3/8), meropenem in high dose or association (3/8), and ceftazidime-avibactam (2/8). Matma [23] reported five neonates with CRE infection treated with colistin in monotherapy with an 80% success rate. Colistin was used in combination with meropenem (75%), amikacin (18.5%), or ciprofloxacin (10.8%) in a study by Eren Çağan on 40 neonates [25]. Ceftazidime-avibactam showed efficacy and tolerability as a salvage therapy in six patients with KPC from a bloodstream infection. [29].

### 3.4. Pseudomonas aeruginosa

We included four articles regarding DTR-*Pseudomonas aeruginosa* (DTR-PA): two case series, one prospective study, and one retrospective study [23,31,32,33]

The mortality, clinical success, and microbiological outcomes are presented in Table 3. All studies were conducted in LMICs. The selected studies report the following infections: BSI, VAP, urine, and others.

Colistin, in association with other antimicrobials, remains the most described antimicrobial for DTR and XDR-PA. However, Belet et al. described the use of ciprofloxacin in 30 infants from a NICU with PA susceptible only to quinolones, with a clinical success of 28 out of 30 [32].

New drugs such as ceftazidime/avibactam or ceftolozane–tazobactam have not been studied in neonates for DTR-PA.

### 3.5. Acinetobacter Baumannii

We identified nine studies about the treatment of XDR *A. baumannii* (XDR-AB): one prospective study, one case report, one case series, and six retrospective studies [23,25,31,33,34,35,36,37,38].

The study characteristics and treatment outcomes are summarized in Table 4. All studies were conducted in LMICs. The selected studies reported as the site of infection BSI, VAP, meningitis, and others.

Colistin is the most represented antimicrobial therapy (8/9).

One case report shows its use for CNS infections by XDR-AB, both intravenous and intraventricular [35], while when used by the intravenous route, it is mainly accompanied by other antimicrobials such as carbapenems and amikacin [23,25,36].

Colistin has shown efficacy as a nebulized therapy for MDR and XDR-AB pneumonia in neonates [37]. Cefoperazone/sulbactam was evaluated in different combinations in XDR-AB VAP in a 12-patient series [41].

### 3.6. Risk of Bias

Data on antimicrobial therapy in children are not conclusive. Our systematic review shows that 31 articles are case reports or case series, and 17 are retrospective studies.

## 4. Discussion

Our research identified twenty-two articles describing the clinical and microbiological outcomes of different treatment options for the most common MDR Gram-negative bacteria in the neonatal population.

Most studies were retrospective and analyzed the use of a specific treatment for different species and resistance patterns, so they were reported in our results more than once (e.g., colistin for CRE and *P. aeruginosa*). To outline clear and comparable results, we included only studies where information about the spectrum of susceptibility, antimicrobial treatment, and outcomes were detailed and quantifiable, excluding descriptive studies on the use of antibiotics in infectious syndromes but without precise microbiological isolation. However, data on the neonatal population are scant, and significant conclusions are challenging to achieve.

ESBL-producing bacteria are the most widespread MDROs, with 10% of colonized neonates becoming infected [42]. Further, in addition to cephalosporins, different *Enterobacterales* species can also share some genetic elements with different patterns susceptible to other antibiotics [43,44].

Extended-spectrum cephalosporins (cefotaxime and ceftriaxone) showed inadequacy in pediatrics, even with low minimum inhibitory concentrations (MICs) in vitro. The same unfavorable response was observed with an aminoglycoside in monotherapy, which is, therefore, discouraged [45].

Carbapenems remain the mainstay of treatment, with most evidence in adults [46,47,48,49] but also a high percentage in pediatrics as well, with evidence of superiority compared to other regimens (piperacillin-tazobactam and amikacin) [50]. In addition, we found a unique article concerning ESBL-producing bacteria in neonates with defined outcomes and treatment in South Africa, describing mortality rates of 25% in the empirical therapy of piperacillin-tazobactam and amikacin and 32% with meropenem; these data must be interpreted with caution since infants receiving meropenem as empiric treatment were more likely to be severely ill [22]. With neonatal sepsis, empirical therapy plays a key role, and local epidemiology must be considered. The NeoMero1 trial in 2020 found no evidence that meropenem as an empiric therapy is superior to the standard of care for neonatal sepsis and that it should be reserved for suspected Gram-negative infections and places with high rates of beta-lactamase circulation [51].

Major challenges come with DTR-*Pseudomonas aeruginosa*, *Acinetobacter baumannii,* and CRE. The role of carbapenems in treating carbapenem-resistant strains remains widely heterogeneous according to the available resources.

A previous systematic review on the treatment of carbapenem-resistant bacterial sepsis reported a fatality rate of 19% for neonates treated with monotherapy, pointing out the paucity of data from which to draw conclusions [52].

We found most studies coming from LMICs, where colistin is the prevalent antimicrobial used [53,54], but the use of carbapenem in association with colistin or other antibiotics is reported, trying to use synergistic or strategic combinations to overcome resistance. However, on the other hand, in HICs more recent antibiotics are being studied.

The Infectious Diseases Society of America (IDSA) 2021 guidance for MDRO treatment recommended colistin only as an alternative strategy if first-line options are not available or tolerated (mainly combination beta-lactamase inhibitors, carbapenems, and monobactam); however, these recommendations are tailored for adults and high-income settings [55].

Colistin (Polymyxin E) has been available for clinical use since the 1950s. The intravenous formulation was gradually rejected worldwide in the early 1980s because of nephrotoxicity. However, in the last two decades, the emergence of XDR organisms and the lack of newer drugs, especially in LMICs, led to reconsidering colistin as a therapeutic option [56].

Colistin has also been used in newborns and preterm infants, appearing to be effective in treating Gram-negative MDROs [57]. Due to its narrow therapeutic index and scant PK/PD studies in children, dosage recommendations for colistin in children and neonates are challenging to achieve [58,59].

Lastly, as a global concern, Gram-negative bacteria can gain resistance to colistin by acquiring a mobilized colistin resistance (*mcr*) gene and other subgroups, which are trending toward worrisome dissemination worldwide [60].

Recently approved beta-lactam agents/beta-lactamase inhibitor combinations (βL-βLI ceftazidime-avibactam, ceftolozane-tazobactam, imipenem-relebactam, and meropenem-vaborbactam) offer activity for CRE, DTR-*Pseudomonas aeruginosa,* and *Acinetobacter baumannii*, largely depending on the resistance mechanism and the subtype of carbapenemase [61]. Avibactam is a β-lactamase inhibitor that is not susceptible to hydrolysis by ESBLs, AmpCs, KPCs, or OXA 48-like carbapenemases. The FDA initially approved it in 2015 to treat complicated UTIs and intra-abdominal infections (IAI) in combination with metronidazole. It exerts no activity against *A. baumannii.* Further studies in HICs revealed its non-inferiority compared to meropenem in pediatric patients and informed the recommended doses. Bradley et al. [62] described ceftazidime-avibactam in combination with metronidazole as non-inferior compared to meropenem in treating intra-abdominal infections in infants older than three months of age, reporting a favorable clinical and microbiological response in ≥90% of cases. The same author reported similar safety and efficacy data on the use of ceftazidime-avibactam in complicated UTIs due to MDR *Enterobacterales* [63]. Ceftolozane-tazobactam has the most potent antipseudomonal activity compared with other βL-βLIs. It was approved in 2014 by the FDA to treat cUTIs and cIAIs in adult patients; Bradley [62,63] showed its efficacy and safety in children in two recent randomized controlled trials for the same indications. Meropenem-vaborbactam consists of an injectable synthetic carbapenem and a boronic acid β-lactamase inhibitor. To date, it has been approved by the FDA to treat adults with CRE-cUTI. [64]. Pediatric data are limited to a single case report in which a KPC-producing *K. pneumoniae* bloodstream infection was treated successfully [65]. A phase 1 study is underway evaluating dosing, PK, and safety in children [66]. We could not find studies about other molecules such as imipenem-relebactam or cefiderocol.

However, these molecules are not yet approved in neonates, and in the literature, few recent case reports are described [27]. Iosifidis reported one case series about using Ceftazidime/avibactam in neonates with a complete success in XDR or PDR *K. pneumoniae* infections [29]. In addition, two case reports (not included in this systematic review because they were published subsequently), showed the efficacy and safety of ceftazidime/avibactam in preterms [67,68].

As already highlighted, the main issue comes with affordability in LMICs. In a recent review, Darlow et al. identified five possible affordable antibiotics for neonatal sepsis in LMICs: amikacin, tobramycin, fosfomycin, flomoxef, and Cefepime [18]. However, their use for MDROs is hindered by widespread resistances in LMICs: 63, 55, 25 and 18% of Gram-negative bacteria (including non-*Enterobacterales*, e.g., *Acinetobacter* species) for tobramycin, cefepime, amikacin, and fosfomycin, respectively [14].

Our research has many limitations: the included studies are heterogeneous in terms of study design and outcome definitions, making it difficult to compare results, excluding the capacity for a meta-analysis. Further, the quality of evidence is low for most studies due to their retrospective nature (or case reports). The resistance mechanism was not always clearly reported, and in many cases, we deduced it from the susceptibility tests available in the article; the microbiological characterization of resistance was performed only in a few studies. Treatments have been evaluated according to antimicrobial choices in our review, but dosages and durations have not been addressed in our research and are pretty different in the single studies, making them even more challenging to describe. Furthermore, most patients included in the studies were critically ill, with multiple comorbidities, and were admitted to an intensive care unit; these factors may lead to underestimating the specific effect of a particular antibiotic treatment on mortality. In addition, the studies were often based on small sample sizes, reducing the ability to find any effect difference and to consider confounder adjustment and multivariate regression analysis.

## 5. Conclusions

Newer drugs are promising against emerging MDROs, but data on the neonatal population are lacking, and more extensive studies are needed. However, older molecules still play a crucial role in LMICs for the treatment of neonatal sepsis, where novel combination regimens must be addressed. Strategies cannot be generalized but will be different according to the country’s epidemiology and resources.

As infections run faster than clinical trials, newer ways to reach evidence in pediatrics need to be arranged, such as using global networks that overcome countries’ differences and real-world-data strategies.

## Figures and Tables

**Figure 1 antibiotics-11-01088-f001:**
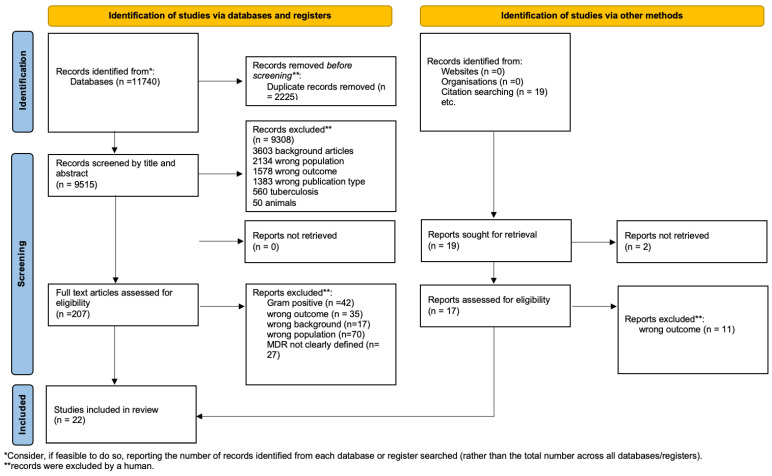
Flowchart of the study selection process.

**Figure 2 antibiotics-11-01088-f002:**
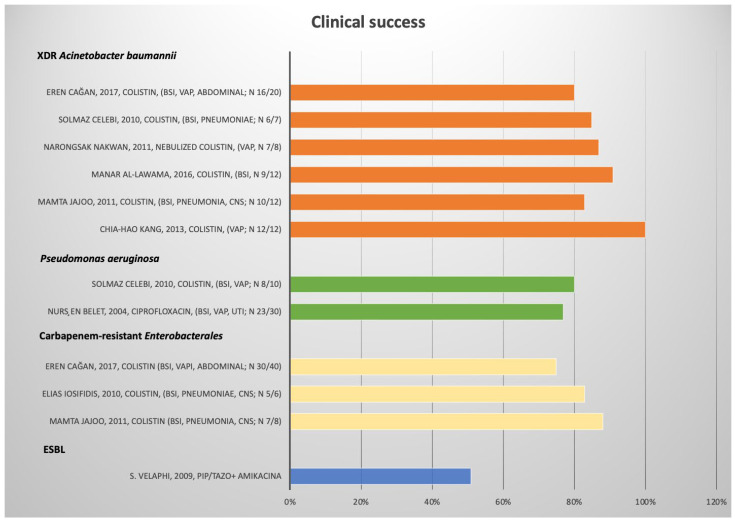
Results regarding choice of treatment and clinical success for ESBL-producing *Enterobacterales* non-UTI infections, Carbapenem-resistant (CR) *Enterobacterales*, *Pseudomonas aeruginosa*, and XDR *A. baumannii*. BSI: blood stream infection; UTI: urinary tract infection; VAP: ventilator-associated pneumonia; CNS: central nervous system.

**Table 1 antibiotics-11-01088-t001:** ESBL Enterobacterales.

Reference	Study Type	Publication Year	Country	Center	Setting	N of Patients (Inc/All)	Median Age (Year)	Resistance	Site of Infection	Antimicrobial Treatment	Route	Evaluated Outcomes	Outcome Measures	Results	Quality Assessment
S. Velaphi [22]	Retrospective	2009	South Africa	Monocenter	NICU	100	neonate	ESBL	BSI, CNS, Lung	Empirical therapy: Pip–Taz + Amikacina vs. Meropenem vs. other vs. none	IV	Mortality	Absolute Value	**Clinical success:** Pip tazo/amikacina: 51%. Meropenem 39%. **Mortality:** 40% (Pip–Tazo + Amikacina); 43% (Meropenem)	fair

NICU: neonatal intensive care unit; ESBL: extended-spectrum β-lactamase; BSI: blood stream infection; CNS: central nervous system; IV: intravenous.

**Table 2 antibiotics-11-01088-t002:** Carbapenem-resistant Enterobacterales.

Reference	Study Type	Publication Year	Country	Center	Setting	N of Patients (Inc/All)	Median Age (Year)	Bacteria	Resistance	Site of Infection	Antimicrobial Treatment	Route	Evaluated Outcomes	Outcome Measures	Results	Quality Assessment
Mamta Jajoo [23]	retrospective	2011	India	monocenter	nicu	5	preterm and term	K	CRE	pneumonia, bsi, cns, chest empyema	Colistin	iv	Clinical success	Absolute value	**Clinical success** 4/5 (80%)	fair
Bonfanti [24]	case report	2016	Italy	monocenter	nicu	1	Preterm	K	CRE	BSI	Colistin	iv	Clinical success	Absolute value	**Clinical success**: 1/1	poor
eren cağan [25]	retrospective	2017	Turkey	monocenter	nicu	40	Preterm	E	CRE	vap, bsi, intrabdominal	Colistin	iv	Clinical success	Absolute value	**Clinical success**: 30/40 (75%)	fair
Escobar Perez JA [26]	case series	2012	Colombia	monocenter	nicu	4	Preterm	K	CRE NDM 1	bsi	Imipenem + Ciprofloxacin; Meropenem + Rifampicin	iv	Clinical success	Absolute value	**Clinical success**: Imipenem + Ciprofloxacin 2/3; Meropenem + Rifampicin 1/1	fair
yesim coskun [27]	Case report	2020	Turkey	monocenter	nicu	1	Preterm	K	PDR	Uti	Ceftazidim/avibactam	iv	Clinical success	Absolute value	**Clinical success**: 1/1	poor
Zhang XY [28]	retrospective	2015	China	monocenter	nicu	8	Preterm	K	CRE NDM 1	Pneumoniae, bsi	Meropenem + ciprofloxacin; ceftazidime, piperacillina/tazobactam + ceftazidime; Meropenem; Meropenem + Piperacillina/tazobactam	iv	Clinical success	Absolute value	**Clinical success**: Meropenem + ciprofloxacin: 0/1; ceftazidime: 2/2; piperacillina/tazobactam + ceftazidime: 0/1, Meropenem: 3/3; Meropenem + Piperacillina/tazobactam 1/1	fair
elias Iosifidis [29]	case series	2019	Greece	monocenter	nicu	6	neonate	K	XDR	bsi	ceftazidime/avibactam	iv	Clinical success; microbiological eradication	Absolute value	**Clinical/Microbiological** success: 6/6	good
Yue-E Wu [30]	case report	2020	China	monocenter	nicu	1	preterm	K	CRE	bsi	Meropenem high dose	iv	Clinical success	Absolute value	**Clinical success**: 1/1	poor

K: *Klebsiaella pneumoniae*; E: *Enterobacterales* spp.; iv: intravenous; bsi: blood stream infection; cns: central nervous system; Uti: urinary tract infection; vap: ventilator-associated pneumonia; XDR: extensively drug-resistant; PDR: pan-drug-resistant; CRE: Carbapenem resistance in Enterobacteriaceae (CRE); NDM: New Delhi metallo-beta-lactamase; nicu: neonatal intensive care unit.

**Table 3 antibiotics-11-01088-t003:** *Pseudomonas aeruginosa*.

Reference	Study Type	Publication Year	Country	Center	Setting	N of Patients (Inc/All)	Median Age (Year)	Resistance	Site of Infection	Antimicrobial Treatment	Route	Evaluated Outcomes	Outcome Measures	Results	Quality Assessment
Mamta Jajoo [23]	retrospective	2011	India	monocenter	nicu	3	preterm and term	MDR	pneumonia, bsi, cns, empyema thoracis	Colistin	iv	Clinical success	Absolute value	**Clinical success** 1/3	fair
İstemi Han Celik [31]	case series	2012	Turkey	monocenter	nicu	1	preterm and term	MDR	VAP	Colistin	aerosolized	Clinical success	Absolute value	**Clinical success:** 1/1	poor
Nursen Belet [32]	case series	2004	Turkey	monocenter	nicu	30	preterm	MDR	VAP, urine, bsi, pleural fluid	Ciprofloxacin	iv	Clinical success	Absolute value	**Clinical success** 23/30 (77%)	fair
Solmaz Celebi [33]	prospective	2010	Turkey	monocenter	inpatient	10	preterm	XDR	VAP, bsi	Colistin	iv	Clinical success	Absolute value	**Clinical success** 8/10 (80%)	fair

nicu: neonatal intensive care unit; MDR: multidrug-resistant; bsi: blood stream infection; cns: central nervous system; VAP: ventilator-associated pneumonia; iv: intravenous; XDR: extensively drug-resistant.

**Table 4 antibiotics-11-01088-t004:** XDR *Acinetobacter baumannii*.

Reference	Study Type	Publication Year	Country	Center	Setting	N of Patients (Inc/All)	Median Age (Year)	Resistance	Site of Infection	Antimicrobial Treatment	Route	Evaluated Outcomes	Outcome Measures	Results	Quality Assessment
Mamta Jajoo [23]	retrospective	2011	India	monocenter	nicu	12	preterm and term	XDR	pneumonia, bsi, cns, empyema thoracis	colistin	iv	Clinical success, mortality	Absolute value	**Clinical success** 10/12 (83%)	fair
eren cağan [25]	retrospective	2017	Turkey	monocenter	nicu	20	preterm	XDR	VAP, bsi, intra-abdominal	colistin	iv	Clinical success, mortality rate	Absolute value	**Clinical success:** 16/20 (80%)	fair
İstemi Han Celik [31]	retrospective	2012	Turkey	monocenter	nicu	2	preterm and term infant	XDR	VAP	colistin	aerosolized	Mortality	Absolute value	**Mortality:** 2/2	poor
Solmaz Celebi [33]	prospective	2010	Turkey	monocenter	inpatient	7	preterm	XDR	pneumoniae, bsi	colistin	iv	Mortality, clinical success	Absolute value	**Clinical success:** 6/7 (85%)	fair
Chia-Hao Kang [34]	case series	2013	China	monocenter	nicu	12	preterm	XDR	VAP	colistin	iv	Clinical success	Absolute value	**Clinical success:** 12/12	poor
Rathna Pratheep [35]	case report	2019	India	monocenter	nicu	1	preterm	XDR	cns	colistin	iv + ivt	Clinical success	Absolute value	**Clinical success:** 1/1	poor
Manar Al-lawama [36]	retrospective	2016	Jordan	monocenter	nicu	21	preterm	XDR	bsi	colistin	iv	Clinical and microbiological eradication	Absolute value	**Clinical success** 19/21 (91%)	good
narongsak nakwan [37]	retrospective	2011	Thailand	monocenter	nicu	8	preterm	XDR	VAP	colistin	aerosolized	Clinical success	Absolute values	**Clinical success:** 7/8 (87%)	good
Thatrimontrichai A [38]	retrospective	2013	Thailand	monocenter	nicu	12	Neonate	XDR	Bsi	ceftazidime; Cefperazone/sulbactam; Imipenem, colistin; Imipenem + cefoperazone/sulbactam; Colistin + Cefoperazone sulbactam	iv	Clinical success	Absolute value	**Clinical success:** ceftazidime 0/1; cefperazone/sulbactam: 2/3; Imipenem: 2/4; Colistin:1/2; Imipenem + cefperazone/sulbactam: 1/1; Colistin + Cefperazone/sulbactam: 1/1	fair

iv: intravenous; ivt: intraventricular; Bsi: blood stream infection; cns: central nervous system; VAP: ventilator-associated pneumonia; nicu: neonatal intensive care unit.

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
