# Peer review of "Therapeutic Options and Outcomes for the Treatment of Neonates and Preterms with Gram-Negative Multidrug-Resistant Bacteria: A Systematic Review"

_antibiotics, 2022, doi:10.3390/antibiotics11081088_

Round 1

Reviewer 1 Report

The authors in this manuscript have reviewed the current treatment regimens and their consequences for the Multi-Drug Resistant Gram-negative bacterial infections in new-born and premature babies. After reviewing the literature reports for the treatment of MDR gram-negative bacterial infections in newborns, the group had concluded that the existing treatment options are playing in important role in containing and treating the infections, however they are not enough to combat drastically emerging MDR Gram-negative bacterial infection. Although the newer potential drugs are promising for treating emerging drug resistant infections, there not enough data points to draw conclusions and more studies are needed for treatment option developments, with main focus on combination therapeutic regimen. In my opinion, the manuscript is well written and I recommend the editors to accept the manuscript as is.

Here are my comments:

1)    Page 2, Line 11: please name the specific classes of broad-spectrum antibiotics the Gram-negative bacteria are susceptible to.

2)    Please introduce the abbreviations before using them. Ex: PICOS, P.A., MEDLINE, MDRO’s.

Author Response

Dear Editors,

Dear Reviewers,

We would like to thank you for the helpful comments and suggestions. We are resubmitting our manuscript after addressing point-by-point all the comments.

Sincerely,

Lorenzo Chiusaroli, on behalf of all the Author

The authors in this manuscript have reviewed the current treatment regimens and their consequences for the Multi-Drug Resistant Gram-negative bacterial infections in new-born and premature babies. After reviewing the literature reports for the treatment of MDR gram-negative bacterial infections in newborns, the group had concluded that the existing treatment options are playing in important role in containing and treating the infections, however they are not enough to combat drastically emerging MDR Gram-negative bacterial infection. Although the newer potential drugs are promising for treating emerging drug resistant infections, there not enough data points to draw conclusions and more studies are needed for treatment option developments, with main focus on combination therapeutic regimen. In my opinion, the manuscript is well written and I recommend the editors to accept the manuscript as is.

We thank the reviewer for the kind words regarding our manuscript.

Here are my comments:

1)    Page 2, Line 11: please name the specific classes of broad-spectrum antibiotics the Gram-negative bacteria are susceptible to

We thank the reviewer for the suggestion, we add the specific classes

2)    Please introduce the abbreviations before using them. Ex: PICOS, P.A., MEDLINE, MDRO’s.

We completely agree with the reviewer’s comment. We spelt out the abbreviations before using them

Reviewer 2 Report

The following are the comments I have for the manuscript entitled: “Therapeutic options and outcomes for the treatment of neonate and preterm with Gram-negative multidrug-resistant bacteria: a systematic review:

The manuscript I downloaded from link website that MDPI Antibiotics sent in the request email does not contain line numbering so I will try to point comments based on sections and reference numbers on the text.

- Grammatical and writing comments:

The manuscript contains lot of acronyms some of them introduced in the Abstract such as MDRO, MDR, HIC or LMIC. They are introduced again in the Background section line with references 4, 5 and 6 but inconsistently. For instance, LMIC is described again (low and middle income countries) but in the same line HIC is mentioned but not introduced.

- In the section Methods/Literature Search, line 4 it reads: “… “neonate” AND “MDR” AND “Outcome assessment”…, It seems that the word AND has been written in uppercase or perhaps is another acronym.

I am not a specialist in modern English language so I may have missed other typos but I would suggest a grammatical review of the text and possibly correction by a native English speaker.

- Content comments:

The authors make a systematic review and mention at least twice (in Literature search and Eligibility criteria sections and partially in the Abstract) that their each is from January 1, 2000 to September 7, 2021, however, the earliest reference showed on tables 1, 2, 3 and 4 is from 2009 (table1). The references for the whole article has the oldest reference from 2002 (reference 44. The next one is 2004 (reference 33). One reference (15) is from 1999 but it seems it might be updated to 2020. The authors mention in the RESULTS, Study selection section that they have excluded 9515 records as well as other hundreds or dozens based on they did not meet eligibility or they regarded gram-positive bacteria. I may have miss it, but are those rejected articles from 2000 to 2002? Or perhaps to 2009 according to table 1? Does it mean that no articles until 2002/2009 show the mater of interest for this review? It is not clear for me so I would suggest authors to mention this situation at least briefly in that Study selection section,

 There are at least 4 references from 2022 (19, 21, 66)  and though not year is shown, reference 67 we may infer it is also 2022 according to the explanation given in DISCUSSION section and 3rd paragraph from the last, where the authors mention these works are not included in the review. No explanation given for inclusion of reference 19.

In all, though the authors recall that “ Our research has many limitations…” (last paragraph in DISCUSSION section), this systematic review could be useful for those working in the field of antimicrobial (gram negative mostly) drugs for newborn children. Considering these limitations and with the suggested clarifications and corrections, this manuscript could be suitable for publication.

Author Response

Dear Editors,

Dear Reviewers,

We would like to thank you for the helpful comments and suggestions. We are resubmitting our manuscript after addressing point-by-point all the comments.

Sincerely,

Lorenzo Chiusaroli, on behalf of all the Author

- Grammatical and writing comments:

The manuscript contains lot of acronyms some of them introduced in the Abstract such as MDRO, MDR, HIC or LMIC. They are introduced again in the Background section line with references 4, 5 and 6 but inconsistently. For instance, LMIC is described again (low and middle income countries) but in the same line HIC is mentioned but not introduced.

We thank the reviewer for the suggestion, We spelt out the abbreviations before using them

- In the section Methods/Literature Search, line 4 it reads: “… “neonate” AND “MDR” AND “Outcome assessment”…, It seems that the word AND has been written in uppercase or perhaps is another acronym.

We are sorry for the confusion. The word AND is written in Uppercase, and it is used as a search connector; we have spelt out all the abbreviations, so in this way, it will be more understandable for the readers.

I am not a specialist in modern English language so I may have missed other typos but I would suggest a grammatical review of the text and possibly correction by a native English speaker.

We submitted the paper to a native English speaker.

- Content comments:

The authors make a systematic review and mention at least twice (in Literature search and Eligibility criteria sections and partially in the Abstract) that their each is from January 1, 2000 to September 7, 2021, however, the earliest reference showed on tables 1, 2, 3 and 4 is from 2009 (table1). The references for the whole article has the oldest reference from 2002 (reference 44. The next one is 2004 (reference 33). One reference (15) is from 1999 but it seems it might be updated to 2020.

Reference 15 refers to update of Surveillance of antimicrobial resistance in Europe from European Centre for Disease Prevention and Control

 The authors mention in the RESULTS, Study selection section that they have excluded 9515 records as well as other hundreds or dozens based on they did not meet eligibility or they regarded gram-positive bacteria.

We thank the reviewer for the suggestion. We excluded Gram-positive bacteria infections from systematic review. We clarified this by adding an appropriate sentence.

 I may have miss it, but are those rejected articles from 2000 to 2002? Or perhaps to 2009 according to table 1? Does it mean that no articles until 2002/2009 show the mater of interest for this review? It is not clear for me so I would suggest authors to mention this situation at least briefly in that Study selection section,

According to the research strategy, we searched every study in the time frame from January 1, 2000, to September 7 2021. Our search has not found any articles from 2000 to 2004 that fit the selection criteria. Therefore, the first article that falls within the selection criteria is dated 2004.

 There are at least 4 references from 2022 (19, 21, 66)  and though not year is shown, reference 67 we may infer it is also 2022 according to the explanation given in DISCUSSION section and 3rd paragraph from the last, where the authors mention these works are not included in the review. No explanation given for inclusion of reference 19.

The bibliography refers to the entire paper, not only to the articles included within the systematic search. We added specific details for reference 67.

In all, though the authors recall that “Our research has many limitations…” (last paragraph in DISCUSSION section), this systematic review could be useful for those working in the field of antimicrobial (gram negative mostly) drugs for newborn children. Considering these limitations and with the suggested clarifications and corrections, this manuscript could be suitable for publication.

We thank the reviewer for the kind words regarding our manuscript.
